# Atherosclerosis in Different Vascular Locations Unbiasedly Approached with Mouse Genetics

**DOI:** 10.3390/genes11121427

**Published:** 2020-11-28

**Authors:** Yukako Kayashima, Nobuyo Maeda-Smithies

**Affiliations:** Department of Pathology and Laboratory Medicine, University of North Carolina at Chapel Hill, CB#7525, 701 Brinkhous-Bullitt Building, Chapel Hill, NC 27599-7525, USA; nobuyo@med.unc.edu

**Keywords:** atherosclerosis, apolipoprotein E-deficient mouse, association study, quantitative trait locus, vascular geometry, endothelial cell, vascular smooth muscle cell, single-cell RNA sequencing

## Abstract

Atherosclerosis in different vascular locations leads to distinct clinical consequences, such as ischemic stroke and myocardial infarction. Genome-wide association studies in humans revealed that genetic loci responsible for carotid plaque and coronary artery disease were not overlapping, suggesting that distinct genetic pathways might be involved for each location. While elevated plasma cholesterol is a common risk factor, plaque development in different vascular beds is influenced by hemodynamics and intrinsic vascular integrity. Despite the limitation of species differences, mouse models provide platforms for unbiased genetic approaches. Mouse strain differences also indicate that susceptibility to atherosclerosis varies, depending on vascular locations, and that the location specificity is genetically controlled. Quantitative trait loci analyses in mice suggested candidate genes, including *Mertk* and *Stab2*, although how each gene affects the location-specific atherosclerosis needs further elucidation. Another unbiased approach of single-cell transcriptome analyses revealed the presence of a small subpopulation of vascular smooth muscle cells (VSMCs), which are “hyper-responsive” to inflammatory stimuli. These cells are likely the previously-reported Sca1^+^ progenitor cells, which can differentiate into multiple lineages in plaques. Further spatiotemporal analyses of the progenitor cells are necessary, since their distribution pattern might be associated with the location-dependent plaque development.

## 1. Introduction

Cardiovascular disease (CVD) is one of the predominant causes of mortality worldwide, resulting from atherosclerotic modifications that compromise the function of blood vessels. Atherosclerosis is a chronic disease of arteries, characterized by stiffness and stenosis of the vessels. It involves multiple vascular beds of primarily the larger arteries, leading to regional clinical manifestations, including coronary heart disease, stroke, abdominal aortic aneurysm, renal failure, and peripheral arterial diseases. In humans, fatty streaks appear in the thoracic and abdominal aortas, early in childhood, and begin to develop in the coronary arteries in the second decade [1]. Raised plaques with complex features increase as individuals become older. In early stages of atherosclerosis, low density lipoprotein (LDL) particles accumulate in the subendothelial space, due to increased vascular permeabilities and transcytosis through vascular endothelial cells (ECs) [2]. Cholesterol-rich LDL undergoes various modifications, including oxidation, and further activates ECs. Activated ECs secrete proinflammatory mediators and upregulate adhesion molecules, enhancing the recruitment of inflammatory cells such as monocytes and T cells. Monocytes differentiate into macrophages in the intima, internalize LDL, transform to lipid-laden foam cells, and amplify inflammation. In response to the inflammatory stimuli, vascular smooth muscle cells (VSMCs) migrate and proliferate, forming a fibrous cap that covers a necrotic core containing foam cell debris, lipids, and cholesterol crystal deposits. Advanced plaques with thickened intima, a large necrotic core and calcification, can erode or rupture, leading to arterial thrombosis and tissue and organ ischemia. Pathogenesis of atherosclerosis is therefore highly complex, involving multiple cell types and processes. As a consequence, although an elevated plasma cholesterol level is the primary contributor to atherosclerosis risk, the disease is significantly accelerated by other health conditions including hypertension, diabetes, and chronic inflammation, as well as lifestyle choices such as smoking, diet, and exercise. Genetic factors are important risk determinants, but they are complex. Unlike diseases caused by a single gene mutation, susceptibility to atherosclerosis in any given individual is determined by various combinations of a number of genetic factors. Each of these factors likely has a small effect, but additively or interactively combined with other factors, including diet and life style, can lead to its full development [3].

While atherosclerotic plaques are formed throughout vessels in the body, there are regional differences in the susceptibility to lesion development. At the local level, susceptible sites for plaque formations are found at branch points of the vascular tree and in the lesser curvature of the aorta, where blood flow is characterized by low shear, oscillatory, and turbulent flow. Clearly, hemodynamic factors play an important role in the development of atherosclerotic plaque, and are extensively researched. Excellent reviews are available on this topic [4,5,6]. At a more global level, while plaques involve primarily the larger arteries, the severity and distribution of plaques vary in vessels, from one location to another, in human patients. This suggests that risk factor profiles for vessels at various locations can be distinct. Although the underlying mechanisms are not fully understood, recent advances in genetics and genome biology proposed a novel concept of “hyper-responsive” subpopulations in the vessels, which might open a way to the targeted prevention and therapies of plaque development. In this paper, we review insights from association studies in humans and mice, and potential mechanisms underlying the location-specific atherosclerosis susceptibilities.

## 2. Susceptibility to Atherosclerosis Varies at Different Aortic Locations

### 2.1. Location-Specific Atherosclerosis in Humans

Since dyslipidemia and adverse environmental factors such as diet are strong risk determinants common to all vascular locations, it is not surprising that there is a strong concordance in the occurrence of plaques from one location to another, in human patients. However, variabilities in the distribution and severity of atherosclerosis were long noted. For example, through careful analyses of severity and distribution of atherosclerosis at autopsy, in patients who died with no evidence of recent or old myocardial infarcts, aortic aneurysms, thrombotic occlusions, or cerebral infarcts, Roberts et al. demonstrated that abdominal aorta is more susceptible to atherosclerosis than thoracic aorta in humans [7,8]. They also noted the relative lack of atheroma in the ascending aorta. The location specific susceptibility was also suggested through transplantation experiments. In a series of transplant studies in dogs, Haimovici et al. implanted athero-prone abdominal aortic segments into the athero-resistant thoracic aorta, or vice versa. They then fed the dogs a high-cholesterol diet to induce atherosclerosis [9,10,11]. Strikingly, the homografts maintained the original susceptibility to plaque formation. Since circulating cholesterol and other factors like inflammatory cytokines are common to all vascular locations, the authors suggested that “susceptibility of the arterial tissue itself rather than its location in the aorta may determine the development and degree of atherosclerosis”.

Epidemiological studies also provided evidence that risk factor profiles for vessels at various locations can be distinct. For example, smoking selectively increased plaques in the abdominal aorta, without influencing the right coronary artery lesions in the 25–34 year old age group in the Pathobiological Determinants of Atherosclerosis in Youth (PDAY) studies [12]. Diabetes significantly accelerated plaques in the lower limbs, while preferential atherosclerosis in the carotid artery was observed in hypertensive patients [13]. Additionally, in a small longitudinal study of 52 patients with proven coronary artery disease, progression or regression of plaques in a specific artery did not parallel changes in other vessels [14]. These observations support that there are regional differences in arterial vessels in susceptibility to atherosclerosis, and that different pathological pathways might be involved in plaque development in vessels at different vascular locations.

### 2.2. Mouse Models of Atherosclerosis

As a model of human atherosclerosis, knockout mice of genes involved in regulation of lipid metabolism, including *apolipoprotein E*-deficient (*Apoe*^−/−^) mice and Low density lipoprotein receptor-deficient (*Ldlr*^−/−^) mice are widely used, as well as high-fat-diet-fed mice, or combinations of these to enhance lesion development. These models develop extensive plaques due to severe dyslipidemia. In these mice, plaques develop first at the aortic root near the attachment sites of the aortic valves, a location where plaques are relatively uncommon in humans. As the mice age, plaques are formed in other vessels, including the aortic arch and innominate artery. Unlike humans, coronary atherosclerosis is not common except in aged mice and in mice with accelerated atherosclerosis, which is induced by diet, additional genetic alterations such as the lack of endothelial nitric oxide synthase [15], or surgical manipulations such as aortic coarctation [16].

Both *Apoe*^−/−^ and *Ldlr*^−/−^ mice develop calcifications in plaques [17,18]. In humans, the sizes, shapes, and locations of calcifications are suggested to be related to plaque vulnerability. Scattered calcified cells and punctate-sign of cell death are observed in early plaques, while large calcification foci are characteristic in the medial layers of advanced atheroma. The calcification process itself is complex, involving multiple factors and pathways, and is under intensive research [19]. Mouse models are useful to study the mechanisms of calcification that cannot be performed in humans, such as lineage tracing studies [20].

While the plaques developing in the aortic tree are human-like, making the mouse models useful for studying the initiation and progression of atherosclerotic plaques, there are limitations. For example, occurrences of plaque rupture are unpredictable incidents in mice [21]. Most plaque ruptures in mice were reported in the brachiocephalic artery of *Apoe*^−/−^ mice, but not in the aortic root or coronary arteries. A majority of the studies were conducted in retrograde using histopathological observations of ruptured plaques. Although live plaque rupture in the right brachiocephalic artery of *Apoe*^−/−^ mice was observed using contrast-enhanced ultrasonography [22], identifying rupture-prone plaques from the histopathology of the plaque remains elusive, and not all plaque ruptures lead to thrombosis [23].

Consequently, most analyses in mice were focused on the early stages, initiation to growth of atherosclerotic plaques.

### 2.3. Strain- and Location-Specific Susceptibility to Atherosclerosis in Mice

Genetic backgrounds of mice influence atherosclerosis. Among the many strains of laboratory mice, C57BL/6 is a standard in studying atherosclerosis, because the strain is susceptible to diet-induced atherosclerosis. Studies using mice on different strain backgrounds revealed that susceptibility to atherosclerosis is different, depending on the mouse strain. In *Apoe*^−/−^ mice of different genetic backgrounds, the lesion sizes in the aortic root areas are ranked as: DBA/2J > C57BL/6J > 129/SV-ter > AKR/J ≈ BALB/cByJ ≈ C3H/HeJ [24]. This order is in general agreement with the finding by Paigen et al. of the diet-induced atherosclerosis susceptibility in wild-type inbred strains of mice [25].

We discovered that susceptibility to atherosclerosis at different aortic locations of mice can vary in a strain-dependent manner, during the characterization of the *Apoe*^−/−^ mice on a 129S6 inbred background (129-*Apoe*^−/−^) [26]. Consistent with earlier observations, *Apoe*^−/−^ mice on the C57BL/6 background (B6-*Apoe*^−/−^) are more susceptible to atherosclerosis than 129-*Apoe*^−/−^ at the root area, despite the lower plasma cholesterol levels in B6-*Apoe*^−/−^ than in 129-*Apoe*^−/−^. Unexpectedly, however, in the aortic arch area, 129-*Apoe*^−/−^ contained much bigger plaques than B6-*Apoe*^−/−^ (Figure 1) [26,27]. In comparison, *Apoe*^−/−^ mice on a DBA/2J background (DBA-*Apoe*^−/−^) are more susceptible to plaque development at both the aortic root and aortic arch than B6-*Apoe*^−/−^. Their plaque sizes were as big as 129-*Apoe*^−/−^, in the aortic arch [28,29].

These observations from the mice on different background strains suggest that responses of vessels to atherogenic stimuli are highly location-dependent, and that the location specificity is genetically determined. These mouse populations on the different genetic backgrounds can be utilized for identification of causative genetic factors through association studies, as discussed in the next section.

## 3. Genetic Studies Support the Presence of Location-Specific Loci for Atherosclerosis

### 3.1. Genome-Wide Association Studies (GWAS) in Humans

Atherogenesis is a highly complex process involving multiple pathways/systems in different organs and tissues. One of the strongest risk factors is elevated plasma cholesterol levels. Low density lipoprotein receptor (LDLR) and apolipoprotein E (APOE) are both crucial for plasma lipoprotein clearance, and LDLR deficiency in humans leads to familial hypercholesterolemia. In addition, a linkage disequilibrium (LD) block containing the 5′ region of the *LDLR* gene was identified by GWAS [30]. This locus is highly predictable for plasma LDL-cholesterol levels, thereby, implicating the level of LDLR transcription as a risk factor. Although a complete lack of APOE is extremely rare in humans, APOE-isoforms with varied receptor binding abilities are common, and are strong predictors of plasma lipid levels. Thus rs4420638, a marker with a strong linkage to *APOE*4*, is associated with higher LDL-cholesterol levels. Approximately 25% of humans carry at least one copy of *APOE*4*.

The genetics of human coronary artery disease, a much more complex condition than increased lipid and cholesterol levels, was tackled by a meta-analysis that compiled GWAS data from a large number of subjects. More than 160 loci reaching statistical significance were detected [31]. While some of the recognized loci were clearly related to lipid metabolism and were likely causative, most were not previously implicated in atherogenesis. A strong association was found in an intergenic region at chromosome (Chr) 9p21, where only a non-protein coding RNA of unknown function (ANRIL) was located [32]. All loci discovered so far, show a modest effect with an odds ratios <1.3. In relevance to the current topic of location-specific susceptibility, another GWAS was performed for carotid intima/media thickness and plaques determined by ultrasonography, both of which are measures of subclinical atherosclerosis [33]. The study identified three loci for carotid intima/media thickness and two loci for carotid plaque, but none were the same as those identified by the GWAS for coronary artery disease. The absence of overlapping factors between these two studies might be because the two studies focused on different clinical stages of atherosclerosis, but it might also be because different genetic factors affect the development of lesions at the two anatomical locations.

### 3.2. Quantitative Trait Loci (QTL) Analysis in Mice

Genomic regions (loci) containing genetic factors underlying the strain-specific differences in complex but quantifiable traits, such as atherosclerotic plaque sizes, could be mapped through genetic analyses of offspring from crosses between the two strains. In contrast to human studies, early-stage plaques are easily detectable in mice, and the environmental factors can be controlled. For this reason, QTL analyses were extensively used for the attempts to identify genes that could modify atherosclerosis, employing typically F2 generations from *Apoe*^−/−^ or *Ldlr*^−/−^ on two susceptible and resistant inbred backgrounds, or from wild-type mice of inbred strain combinations that were fed an atherogenic diet. More than 50 loci were reported, and several strong candidate genes were identified, including *Tnfsf4* on Chr 1 [34], *Rcn2* on Chr 9 [35], *Tnfaip3* on Chr 10 [36], and *Adam17* on Chr 12 [37], although details of the functional consequences of the variations in each candidate gene are not available [reviewed in [38]]. Furthermore, we note that most studies were focused on plaques developing at the aortic root, and information regarding the effects of these QTLs on other vascular beds is not known.

As in human GWAS studies, there are limitations in mouse QTL analysis. One of the obstacles is that, the identified locus tends to be broad and usually contains many genes. Another difficulty is that linkage studies in multi-factorial diseases, such as atherosclerosis, generally explain only part of the variations. This makes proving the causative relationships difficult, because phenotypic differences caused by a given gene variant can be too subtle to detect, particularly when the phenotype in question is the histological assessment of plaque size. Moreover, there is currently no effective way to analyze gene-gene interactions that might account for a considerable part of the phenotype variations. Epistatic interactions can hamper the identification of causative genes, when multiple genes are clustered in the locus, and even interactions between genes on different chromosomes are not rare. Identification of genes in a given QTL requires heroic efforts, as illustrated in a review by Rapp and Joe on the QTL analysis of blood pressure regulation in rats [39]. Despite these difficulties, information on genomic structure and function is increasing very quickly, giving us useful tools to narrow down the QTL regions and determine the underlying genes.

### 3.3. QTLs for the Location-Specific Atherosclerosis

The observation that plaque development in mice can differ in location- and strain-specific manners, as described in Section 2.3. above, led us to embark on QTL analyses. In an effort to minimize pitfalls we might encounter due to potentially profound genetic background effects on each QTL, we took a novel approach of analyzing three sets of F2 populations from crosses of *Apoe*^−/−^ mice on the C57BL/6, 129S6, and DBA/2J backgrounds, in a circular fashion (Figure 2A) [27,28,29,40]. As every genetic variant was examined at least two times in the round-robin design of the three crosses, a given QTL with large additive effects and strong penetrance is expected to be detectable in two out of the three crosses, if no epistatic background effects are present.

QTL analysis using each set of F2 generation *Apoe*^−/−^ mice from C57BL/6 × 129S6, DBA/2J × 129S6 and C57BL/6 × DBA/2J crosses and maintained on regular mouse chow (4.5% fat, 0.02% cholesterol), identified new QTLs for the arch lesion size (*Aath1-5*, Table 1) [27,28,29,40]. *Aath4*, on Chr 2 at 136 Mb, was detected by significant QTL peaks in DBA/2J × 129S6 and C57BL/6 × DBA/2J but not in C57BL/6 × 129S6 (Figure 2B), suggesting that genetic variants unique to DBA/2J at *Aath4* confer susceptibility to atherosclerosis at the aortic arch. The analyses also detected *Aath5* on Chr 10: 60 Mb in crosses DBA/2J × 129S6 and C57BL/6 × DBA/2J but not in C57BL/6 × 129S6. In this case, the variants unique to DBA/2J at *Aath5* protect against the arch plaque development. Some of the identified loci including *Aath1* (Chr 1:109 Mb) and *Aath2* (Chr 1:163 Mb) showed significant peaks in only one of the three crosses (C57BL/6 × 129S6), likely due to epistasis, although the absence of statistical power cannot be eliminated. Importantly, the arch QTLs *Aath1-3* and *Aath5* did not overlap with QTLs for root lesion size, supporting the idea that distinct genetic factors are involved in atherogenesis at each location (Table 1). An exception is *Aath4* on Chr 2, whose peak overlaps with *Athla1* for the aortic root lesion size, which was identified by a cross between *Ldlr*^−/−^ mice on the C57BL/6 and PERA/Ei backgrounds [41]. *Ath45* also overlaps with *Athla1*, but the allelic effects in *Ath45* are not consistent among the three crosses, suggesting that multiple loci with opposing effects are present in this region, affecting the aortic root plaque development.

### 3.4. From QTL to the Gene

By searching genome sequences in the *Aath4* region where DBA/2J is unique, while C57BL/6 and 129S6 share the variants, we identified Mer proto-oncogene tyrosine kinase (*Mertk*) as a candidate gene for *Aath4*. *Mertk* encodes a phagocytosis receptor for apoptotic cells and plays an important role during atherosclerotic plaque development. *Mertk*^−/−^*Apoe*^−/−^ mice showed accumulation of apoptotic cells and expansion of necrotic cores within plaques [42], and *Ldlr*^−/−^ mice transplanted with *Mertk*^−/−^ bone marrow, showed accumulation of apoptotic cells and accelerated atherosclerosis [43]. A congenic mouse line *Aath4^DBA/DBA^*, in which the DBA/2J allele of *Aath4* was backcrossed onto 129-*Apoe*^−/−^, showed a larger plaque size than the control, confirming that the DBA/2J allele of *Aath4* confers susceptibility to atherosclerosis. In macrophages from *Aath4^DBA/DBA^*, expression of *Mertk* was reduced by 50%, and phagocytosis of apoptotic cells was reduced, suggesting *Mertk* is a contributing gene for *Aath4* [44]. While there are nine amino acid alterations in the MERTK protein of DBA/2J compared to 129S6, these two types of MERTK did not show significant differences in the efficiency of phagocytosis. Therefore, reduced transcription of *Mertk*, but not differences in the MERTK protein structure, likely contributes to the phenotype. Reduced efferocytosis is consistent with the mechanism of the DBA/2J-specific reduction in *Mertk* gene expression and the reduced expression of *Mertk* associates with increased plaques, both in the arch and root of mice. Role of *Mertk* gene expression in the arch-specific atherosclerosis requires further studies, including searching other genes within *Aath4*.

A candidate gene underlying *Aath5* is Stabilin 2 (*Stab2*), which encodes a clearance receptor for hyaluronan. Hyaluronans are large polymers of alternate D-glucuronic acid and N-acetyl-D-glucosamine residues repeating more than thousands of times, which provide increased viscoelasticity to tissue fluids and the cellular microenvironment. DBA/2J mice show more than 10 times higher plasma concentration of hyaluronans than C57BL/6 and 129S6 mice. Consistently, in the liver sinusoidal endothelial cells (LSECs), where *Stab2* is most abundantly expressed, mRNA levels of *Stab2* in DBA/2J mice are less than 30%, compared to the levels in C57BL/6 and 129S6. At the promoter region of *Stab2*, there is a DBA/2J-specific insertion of a retrovirus-derived transposable element, intracisternal A particle (IAP), which likely interferes with normal transcription in the LSECs in DBA/2J. On the other hand, the *Stab2*-IAP element drives ectopic transcription of *Stab2* in tissues where it is not normally expressed. The ectopic expression is dominantly suppressed by loci on Chr 13: 59.7–73.0 Mb from C57BL/6J. The loci encode a large number of genes encoding Krüppel-associated box-domain zinc-finger proteins, which target transposable element-derived sequences and repress their expression. While the same region in 129S6 also show dominant inhibitory effects on the ectopic expression of *Stab2*, additional loci are required for complete suppression [45]. Since hyaluronan is one of the major components of endothelial glycocalyx, and its integrity is important to protect vessels from injury, a reduced plasma hyaluronan clearance could modify vessel response to atherosclerotic stimuli. This is consistent with the observations that the administration of hyaluronans significantly inhibited arterial cell proliferation in injured arteries [46,47]. On the other hand, degraded hyaluronan within the plaque was reported as a mediator of inflammation [48]. Despite the support from these observations, whether and how *Stab2* relates to the location-specific atherosclerosis remains to be elucidated.

### 3.5. Genetic Analyses of Other Mouse Populations

Shi et al. studied plaques in left common carotid artery in F2 mice from intercrosses of *Apoe*^−/−^ mice on B6 × C3H/HeJ, B6 × Balb/cJ, and Balb/cJ × SM/J, fed a Western-type diet (21% fat, 34% sucrose, and 0.15% cholesterol). The authors found a large fraction of F2 mice had little or no plaques in their carotid arteries, despite being severely hyperlipidemic. Nevertheless, analysis of combined data on carotid atherosclerosis from these three crosses identified four highly significant loci, *Cath2* and *Cath7* on Chr 5, and *Cath1* and *Cath3* on Chr 12 and Chr 13, respectively [49]. None of these were found in the QTL analyses of arch or root of crosses involving 129S6, C57BL/6J, and DBA2/J. Some candidate genes for *Cath1* on Chr 12 were suggested, but whether or not *Cath1* is unique to plaques in carotid arteries or overlaps with other QTLs for aortic root plaques in the same cohorts were not discussed. The functional relevance of these variations to atherosclerosis and to location-specificity requires further studies.

QTL analysis of F2 animals derived from two parental inbred strains depends on chromosomal recombination. This limits the resolution power in narrowing the region spanning a given QTL, partly because the recombination frequency along each chromosome is not uniform and there are hot spots as well as cold spots. To overcome this limitation, multiple hybrid populations were developed in which high-resolution genetic mapping could be performed. For example, the Collaborative Cross (CC) strains were made by intercrossing eight inbred strains of mice—A/J, C57BL6/J, 129S1/SvImJ, NOD/ShiLtJ, NZO/HiLtJ, CAST/EiJ, PWK/PhJ, and WSB/EiJ [50]. In the CC strains, founder strains contribute equally to each line. The Diversity Outbred (DO) mouse population was also developed from the same founder strains as CC but were maintained by randomized outbreeding [51]. As a consequence, each DO mouse has a high allelic heterozygosity, and the DO mouse populations provide useful platforms for mammalian system genetics [52]. QTL analysis using the DO population provide high-resolution, due to dense recombinations, high genetic diversity, and genetic randomization. Finally, the Hybrid Mouse Diversity Panel (HMDP) consists of 30 classical inbred strains and 70 or more recombinant inbred strains of C57BL/6J and DBA/2J (the B **×** D RI set) and A/J and C57BL/6J (the A **×** B and B **×** A RI sets) [53]. These mice are highly diverse in metabolic and cardiovascular traits, relevant to human disease [54]. Association analyses utilizing the HMDP panel revealed a number of loci associated with atherosclerosis and relevant traits, such as plasma lipid levels, metabolites and cytokines, with high resolution [55].

Despite a high promise that these mouse populations would make complex genetic traits tractable and disease-susceptible genes identifiable, their utilization has just begun. An eQTL study using DO mice fed a high-fat cholesterol-containing diet was reported. The authors identified a locus on Chr 6 for atherosclerosis at aortic root, and suggested *Apobec1*, an apolipoprotein B mRNA-editing enzyme, as a candidate gene [56].

## 4. What Causes the Location-Specific Susceptibility to Atherosclerosis?

### 4.1. Vascular Geometry, Hemodynamics, and Atherosclerosis

There are variations in the individual geometry of the aorta in humans, and computational studies support the idea that hemodynamics in the aorta is highly dependent on the geometry [57,58,59]. Similarly, there are significant strain-specific differences in mice in the geometry of the aortic arch, such as diameter of aorta, degree of arch curvature, and distances between branches [27,60]. Genetic differences influence vascular geometry and hemodynamics, which could contribute to the distinct susceptibility to plaque development at the aortic arch. Indeed, three-dimensional geometry of the aortic arch obtained by light stereo-microscopic imaging revealed that athero-resistant C57BL/6J mice have a larger arch diameter than athero-susceptible 129S6 mice, and that the aortic arch of C57BL/6J mice is more planar than that of 129S6 mice. By computational fluid dynamic calculations, it was estimated that the 129/SvEv strain had lower shear stress at the inner curvature of the aortic arch, compared to the C57BL/6 strain [60]. However, Doppler ultrasound measurements in vivo showed that the mean relative wall shear stress over the aortic arch in 129-*Apo**e*^−/−^ and B6-*Apoe*^−/−^ mice were not different. While arch plaque sizes negatively correlated with mean relative wall shear stress of individual animals, additional factors are necessary to account for the strain differences in the susceptibility for the arch plaque development [61].

In the F2 populations from the crosses of 129-*Apoe*^−/−^ and B6-*Apoe*^−/−^ described above in Section 3.3., a significant QTL peak conferring the shape of arch curvature, overlapped with a QTL for aortic arch plaque, *Aath1* on Chr 1 [27]. However, QTL was not detectable in the two remaining crosses between B6-*Apoe*^−/−^ and DBA*-Apoe*^−/−^ or 129-*Apoe*^−/−^ and DBA*-Apoe*^−/−^, suggesting that although the determination of the arch curvature is genetically controlled, it also involves multiple interactive factors. The overlapping region in *Aath1* contains many candidate genes including those involved in the developmental tissue patterning. *Gli2*, a component of Hedgehog (Hh) signaling, is one of the candidate genes. Although a recent study showed that Hh signaling is not required for vascular development in mammals [62], *Gli2* is involved in the regulation of ciliary length in fibroblasts in culture [63]. Additionally, a high hedgehog signaling activity was also detected during the perinatal stages of mice, which was restricted to the adventitia of the artery wall that supports the vascular progenitor cells [64]. While the signaling activity diminishes in adult vessels, its role in response to tissue injury is a tantalizing possibility. Of note, Arabani et al. recently reported reductions of atherosclerosis in both *Apoe*^−/−^ and *Ldlr*^−/−^ mice by inactivation of the *Hhipl1* gene, coding for hedgehog interacting protein-like 1 and a candidate gene for coronary artery disease, identified in human GWAS studies [65].

### 4.2. Developmental Origin of Vascular SMCs and Atherosclerosis

The homograft experiments described in Section 2.1. above suggested that blood vessels at different aortic locations can have intrinsic characteristics sufficient to maintain their responses to the shared blood abnormalities. This possibility is supported by indisputable evidence that ECs, VSMCs, and adventitial cells at different locations of aorta have intrinsic differences. SMCs from different vascular beds show variations in morphology, gene expression patterns, and responses to inflammatory stimuli [66,67]. The heterogeneity might be partly a result of different vascular geometry and hemodynamics, but might also be due to intrinsic factors, such as the developmental origin of each vascular bed [68]. Thus, experiments such as fate mapping and lineage tracing of cells during development, clearly defines that SMCs in ascending aorta and aortic arch, brachiocephalic, subclavian, and carotids arteries, originate from neural crest (embryonic neuroectoderm, NE), while the aortic root is derived from secondary heart field (lateral plate mesoderm, LM), and the descending aorta is derived from somites (paraxial mesoderm, PM) (Figure 3). The differences in the developmental origin could cause the distinctive traits of SMCs in adults, such as responses to external stimuli. For example, Topouzis and Majesky demonstrated that the primary culture of SMCs isolated from arotic arch (AA) and from descending abdominal aorta (DA) of chick embryos were similar in many criteria but dramatically differed in TGFβ1-stimulated cell proliferation. They found that TGFBR2, one of the three receptors for TGFβ1, is not glycosylated in SMCs from AA but is fully glycosylated in SMCs from DA [69].

To further elucidate whether and how the developmental origins of SMC could influence the development of vascular disease, Cheung et al. initially induced human pluripotent stem cells (hESCs) into the three embryonic cells, NE, LM, and PM, which they further induced to SMCs that display contractile ability in response to vasoconstrictors [71]. Gene expression profiling of three subtypes showed shared upregulation of 3604 genes, compared to undifferentiated human embryonic stem cells. Yet, these SMC subtypes displayed different behavior in the same way as that of aortic SMCs of distinct origins, and displayed lineage-dependent proliferation differences in response to cytokine stimulation. Authors further described that LM-SMCs showed the highest activity in the degradation of extracellular matrix, when exposed to IL-1β, which shows the highest induction of the *Mmp9* gene expression and the lowest of *Timp1*. PM-SMC was resistant to extracellular matrix degradation with the highest response of *Timp1* gene induction and lowest in *Mmp9*. These results strongly suggest that SMCs at each vascular bed keep their “embryonic memories”.

On the other hand, comparison of aortic and coronary SMCs cultured in different degrees of stiffness of matrix suggested that the transcriptome of coronary SMCs cultured in the stiff matrix is more similar to aortic SMCs cultured in stiff matrix, rather than the coronary SMCs cultured in soft matrix, indicating that microenvironment such as ECM stiffness might override the embryological lineage contributions to the transcriptome [72].

### 4.3. Intrinsic Heterogeneities of Vascular Cells Revealed by Whole Genome Transcriptome Analyses

Recent advances in whole genome transcriptome analysis began to provide evidence and define the intrinsic differences of global gene expressions at different vascular locations. For example, microarray data comparing gene expressions in the descending thoracic aorta (TA) and the aortic arch (AA) revealed that 219 genes are differentially expressed between TA and AA, in both wild-type and *Apoe*^−/−^ mice [73]. Among them, *Homeobox 6* to *10* (*Hox6*-*10*) are highly expressed in TA compared to AA. Location-specific differences in the *Hox* gene expression levels could simply be the consequences of “hard-wired embryonic programs and not postnatal cues”. Nevertheless, the authors further showed that *Hox**a9* contributes to a lower activity of the proinflammatory and proatherogenic nuclear factor-κB (NF-κB) in SMCs isolated from TA, suggesting its potential contribution to the relative atherosclerosis resistance of TA. While this study focused on the genes that differ between TA and AA commonly in the *Apoe*^−/−^ mice and the wild-type mice, it also identified 122 genes and 27 genes that differ uniquely in the *Apoe*^−/−^ mice and the wild-type mice, respectively, which requires further attention.

At a single-cell level, Kaur et al. reported heterogeneity in the expression pattern of G protein coupled receptors (GPCRs) in primary vascular SMC and EC from different vascular beds [74]. In approximately 60 individual SMCs freshly isolated from the aorta (SMao) and from skeletal muscle arteries (SMsk) of 7–8 healthy adult mice, the expression patterns of 154 GPCRs examined in the two SMC types were significantly different. Seven GPCRs were highly expressed in SMao, while most of the other GPCRs were mainly expressed in SMsk. ECs from lung, skeletal muscle, or the brain, also showed a highly heterogeneous pattern of the GPCR expression. The authors detected a total of 76 GPCRs in 60 SMao from 8 healthy adult mice, but only 8 GPCRs were expressed in more than 90% of cells and 19 GPCRs were expressed in more than 50% of cells. Moreover, a small subpopulation of dedifferentiated SMCs was also identified in healthy aorta. The dedifferentiated phenotype was associated with the expression patterns of genes indicating inflammatory activation, as well as increased expression of GPCRs, including an orphan receptor, *Gprc5b*. Immunohistochemical analysis of the *Gprc5b*-*βgal* reporter mice indicated that the dedifferentiating SMCs were enriched in the atheroprone inner curvature of the aorta. Although the number of cells analyzed were limited, the study strongly suggests that some cells are prone to respond more strongly to atherogenic stimuli than others. Further studies with a higher number of cells would be required to understand the detailed profiles of individual cells in different vascular locations.

Surprisingly a high level of heterogeneity at single cell levels was also revealed in the transcriptome comparisons with single-cell RNA sequencing by Dobnikar et al., who provided comprehensive expression analyses in cells between different vascular regions, cells within a specific region, and within a subpopulation of cells [75]. In bulk RNA sequencing profiling of vascular SMCs, 88 genes differentially expressed between the aortic arch (AA) and descending thoracic aorta (TA) of healthy mice were identified as region-specific. Expression of the *Hox* genes was enriched in TA, while genes related to immune response, cell proliferation, and migration were enriched in AA. At a single-cell level, some of the genes, including *Hand2* and *Pde1c*, were almost exclusively expressed in AA, while *Hoxa7* was almost exclusively in TA. However, many genes showing region-specific expression levels in the bulk analysis were expressed at similar levels in the cells from AA and TA, while the number of expressing cells was enriched in either AA or TA. This raises a possibility that the region-specific gene expression profile is caused by the different composition of cells expressing these genes in each region. However, clustering of signature genes suggested that the features of regional identity are borne by individual cells. Within VSMCs in TA and in AA, authors found highly variably expressed genes that segregated the *Myh11*-expressing SMCs into seven clusters. Many of the genes have documented roles in cardiovascular disease, inflammation, and VSMC phenotype, suggesting the functional heterogeneity of VSMCs.

### 4.4. SMC Plasticity and Residential Vascular Progenitor Cells

Single-cell RNA sequencing by Dobnikar et al. also demonstrated that the rare population of SMCs that express Sca1 (Stem cell antigen 1) are present both in healthy as well as in atherosclerotic vessels, and the cells isolated from a specific region of the vessel are also highly heterogeneous in their gene expression patterns [75]. The Sca1-positive cells generally showed a lower expression level of genes linked to contractile vascular SMC, such as muscle contraction, actin filament organization, and cell-substrate adhesion. On the other hand, these cells expressed higher levels of genes associated with migration, proliferation, and secretion of the extracellular matrix, as well as activation of the signaling pathways, reflecting a responsive cell state. When they cultured the Sca1-GFP negative medial cells isolated from Sca1-GFP animals, induction of GFP expression was observed in 15–28% of the cells in more than 7 days of culture. Using a vascular injury model, they further demonstrated that Sca1 expression was induced in 10–45% of VSMCs from the carotid arteries, 8 days after carotid ligation. These experiments suggest that Sca1 expression in VSMCs results from the upregulation of Sca1 transcription in VSMCs upon phenotypic switching, rather than selective expansion of pre-existing Sca1^+^ cells.

It was long recognized that VSMCs have plasticity, which contributes to atherosclerosis. SMCs in the medial layers are phenotypically contractile, but once adapted to the tissue culture in vitro as well as in response to external stimuli, they can switch to a “synthetic” phenotype, acquiring a gene expression pattern for increased cell proliferation, much like cells during development. This plasticity is fundamental to the vascular cell response to injury. More recently, it was shown that vessels also harbor small populations of cells that are capable of differentiating to mature vascular cells. The first clear demonstration of progenitor cells residing in media or adventitia of adult vessels that can differentiate into mature SMCs or ECs was made by Hu et al. [76]. The SMC progenitors are Sca1^+^ cKit^low^ CD34^−^ and have the capability to integrate into the neointimal population after vein graft transplantation. These progenitor cells did not originate from the circulating bone marrow-derived cells but reside within the vasculature. Shortly after, a Sca1^+^ cKit^low^ CD34^low^ Lin^−^ side population of vascular progenitor cells was isolated from the medial layers, and the CD34^+^CD31^−^ cells were isolated from immediately outside of the medial SMC layer [77,78]. These vascular progenitor cells are capable of differentiating into SMCs and ECs, and contribute to vascular remodeling. The origin of these progenitor cells is under investigation. They could be residential undifferentiated cells of developmental origin, or fully matured differentiated cells that dedifferentiate and begin to re-express the stemness markers. Majesky et al. demonstrated that differentiated SMCs generate a subpopulation of Sca1^+^ progenitor in the adventitia, and the process was regulated by Klf4 [79]. Other lineage tracing studies also showed that VSMCs within the plaque are derived from a subset of differentiated, highly plastic VSMCs [80,81].

What is remarkable about the single cell analyses by Dobnikar et al. is that they have demonstrated that the Sca1^+^ VSMC lineage cells are also heterogeneous, and that Sca1-positive cells constitute a significant proportion of cells in atherosclerotic plaques of *Apoe*^−/−^ mice, in contrast to low frequency in healthy tissue, proposing that Sca1 expression is the hallmark of VSMCs undergoing phenotypic switching [75]. Among the VSMC-lineage cells marked by the *Myh11*-driven multicolor Confetti reporter in the plaques, there were also cells that were low in contractile gene expression, yet were expressing chondrocyte-specific or macrophage-specific genes. Thus, phenotypic changes of the VSMC lineage cells in atherosclerotic plaques are dynamic, and have much broader plasticity than originally thought.

Together, the unbiased, high-throughput approach is promising, although spatiotemporal analysis of the dedifferentiation and re-differentiation of the specific subpopulation is critical. These studies raise further questions: Do the cells under the “hyper-responsive” state also exist in the aortic root and coronary arteries? Are there any strain-specific differences in the localization and frequency of the subpopulation? Comparison between the aortic arch and root using different mouse strains will give us the clue. Whether the subpopulation of athero-prone cells exists in humans is another important question, since these subpopulations could be the selective target to prevent and treat the progression of atherosclerosis.

## 5. Conclusions

There is a remarkable technical progress in acquisition of large data and their analyses. Genetic association studies in both humans and mice raised novel candidate genes and pathways underlying location-specific atherosclerosis in an unbiased fashion. However, validation of these candidate genes remains technically challenging, partly because the effect of each locus is small. Plaque development is a complex process involving multiple genes and pathways, and small variations in environmental factors such as nutrients could influence it. As association studies depend on the presence of genetic variations, results obtained in mice and in humans do not always overlap. Moreover, mouse models of atherosclerosis have a serious limitation in studying “vulnerable” plaques in humans, which lead to clinical consequences such as myocardial infarction and stroke. Nevertheless, there are large overlaps in the pathological processes of plaque development in mice and humans, and mouse models remain an important tool in validating the nature of gene–gene and gene–environment interactions. Future challenges would be to increase the data size and also the development of new approaches to apply the genetic data for pathogenesis of complex diseases, such as atherosclerosis.

Advances in other unbiased approaches include single-cell transcriptome analyses, the results of which suggested the heterogeneity of cells in different vasculatures of mice, and the presence of a small number of VSMCs, which are “hyper-responsive” to inflammatory stimuli. These cells are likely the previously-reported Sca1^+^ progenitor cells, which can differentiate into multiple lineages in plaques. It raises a possibility that the uneven distribution patterns of these hyper-responsive cells within vasculature or individuals might contribute to the location- and strain-specific susceptibility. Alternatively, it could be a stochastic response of vascular cells to atherogenic stimuli. Further spatiotemporal analyses of the progenitor cells using mouse models should shed light on these aspects. Whether the progenitor cells exist in the human vasculature also needs attention, since these cells could be a therapeutic target of atherosclerosis at an early stage in humans.

## Figures and Tables

**Figure 1 genes-11-01427-f001:**
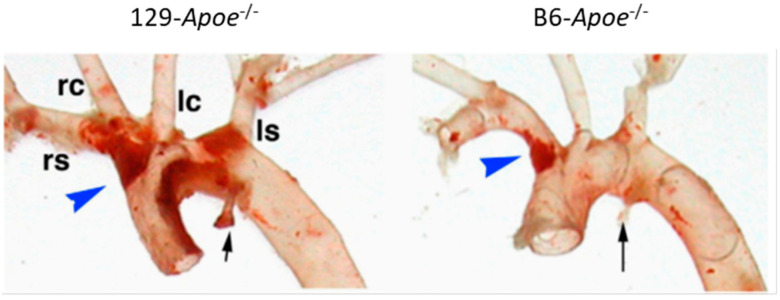
Strain- and location-specific atherosclerosis susceptibility. Plaques in the aortic arches of *Apoe*^−/−^ mice on the genetic background of 129 (left panel) and C57BL/B6 (right panel) mice stained red with Sudan IVB. The brachiocephalic trunks (blue arrowheads) are shorter in 129 than in B6. Black arrow, the attachment site of the ductus arteriosus to each aorta; rs, right subclavian artery; rc, right common carotid artery; lc, left common carotid artery; ls, left subclavian artery. Modified from [26].

**Figure 2 genes-11-01427-f002:**
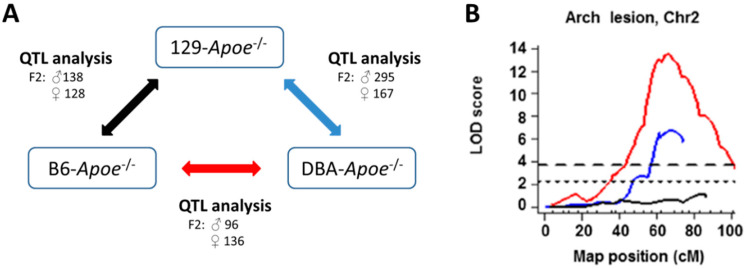
Three-way QTL analyses. (**A**)**.** F2 populations of intercrosses between B6-*Apoe*^−/−^, 129-*Apoe*^−/−^ and DBA-*Apoe*^−/−^ mice were used for the three-way QTL analyses. (**B**)**.** LOD curves of *Aath4* on Chr 2. LOD curves of a cross between B6-*Apoe*^−/−^ and DBA-*Apoe*^−/−^ (red line), a cross between DBA-*Apoe*^−/−^ and 129-*Apoe*^−/−^ mice (blue line), and a cross between B6-*Apoe*^−/−^ and 129- *Apoe*^−/−^ mice (black line) on Chr 2. The horizontal dashed and dotted lines represent thresholds for suggestive QTL (*p* = 0.63) and significant QTL (*p* = 0.05) determined in single locus scan of the cross between B6-*Apoe*^−/−^ × DBA-*Apoe*^−/−^ mice. The significance thresholds for LOD scores were determined by 1000 permutations using R/qtl software. Adopted from [29].

**Figure 3 genes-11-01427-f003:**
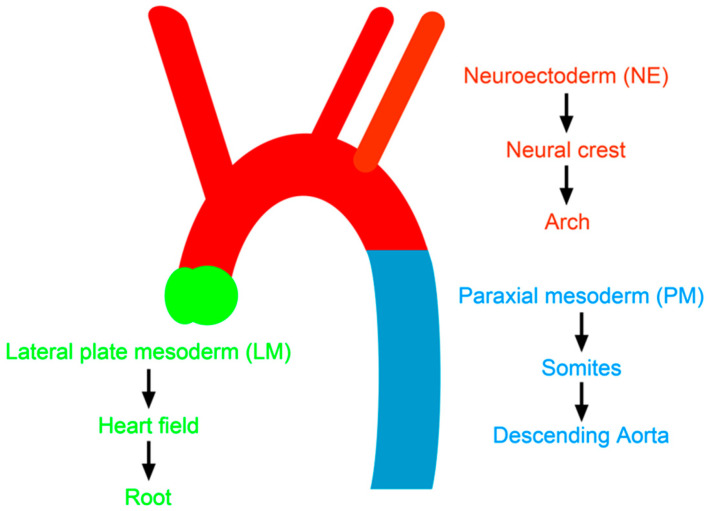
Developmental origin of aorta and strain-dependent susceptibility to atherosclerosis. Aortic arch, root and descending aorta originate from neuroectoderm (NE), lateral plate mesoderm (LM) and paraxial mesoderm (PM), respectively. Modified from [70].

**Table 1 genes-11-01427-t001:** Significant QTLs for atherosclerotic lesions in three F2 intercrosses of *Apoe*^−/−^ mice on B6, DBA, and 129 backgrounds.

Phenotype	QTL	Chr	Peak Mb (CI)	B6 × 129	DBA × 129	B6 × DBA	Consistency
Arch lesion size	*Aath1*	1	109 (85–135)	129 > B6			no
Arch lesion size	*Aath2*	1	163 (151–173)	129 > B6			no
Arch lesion size	*Aath4*	2	136 (124–147)		DBA > 129	DBA > B6	129 = B6 ≠ DBA
Arch lesion size	*Aath5*	10	60 (38–100)		129 > DBA	B6 > DBA	129 = B6 ≠ DBA
Arch lesion size	*Aath3*	15	91.8 (80–102)	129 > B6			no
Root lesion size	*Ath44*	1	158 (153–168)		DBA > 129		no
Root lesion size	*Ath45*	2	162 (154–165)	B6 > 129	DBA > 129	DBA > B6	multiple
Root lesion size	*Ath29*	9	61 (47–71)	B6 > 129			no
Root lesion size	*Ath31*	7	78 (55–84)		129 > DBA	B6 > DBA	129 = B6 ≠ DBA

QTL, quantitative trait locus; Chr, chromosome; CI, 95% credible interval. > shows allelic effects in F2 mice. For example, 129 > B6 indicates that F2 mice homozygous for 129 allele at the locus develop larger plaque than those with B6 allele.

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
