# Peer review of "Atherosclerosis in Different Vascular Locations Unbiasedly Approached with Mouse Genetics"

_genes, 2020, doi:10.3390/genes11121427_

Round 1
Reviewer 1 Report
The work was done at a good methodological level. In my opinion, the study will be of interest to a wide range of researchers. A small recommendation to revise the conclusions of the article, the conclusions do not reflect the significance of the study and do not emphasize the novelty of the results.
The review is devoted to the analysis of the causes of atherosclerotic lesions in mice and humans. The problem of stress shear, the presence of a special population of cells, and concomitant diseases are considered. The emphasis is on the genetic aspects leading to the development of atherosclerosis.
Section abstract: it should be clarified which pathways and/or genes are responsible for developing atherosclerotic lesions; in humans or mice? What is known about human genetic characteristics? There is only one sentence about people. In general, the abstract should be better structured.
The findings do not reflect the entirety of the content: for example, "Susceptibility to atherosclerosis is different depending on vascular beds, and the difference is determined by the combination of the local environment and intrinsic vascular integrity, both of which
are influenced by genetic variance. "the phrase is very vague, it is worth specifying what is meant and what has been clarified.
"Accumulating evidence supports the presence of the subpopulation of VSMCs, which are hyper-responsive to inflammatory stimuli. It is tempting to hypothesize that the distribution patterns of these hyper-responsive cells depending on vascular locations and individuals may contribute to the location- and strain-specific susceptibility. "- it is not clear what these cells are? in humans or mice? What else is known about them? Is this a personal assumption of the authors?
"This knowledge derived from mice could lead us to understand pathological processes involved in the location-
dependent susceptibility to plaque development in humans. " — If this is the main conclusion of the article, then it is not powerful enough and does not show novelty.
Reviewer 2 Report
Manuscript review for Genes November 2020
Atherosclerosis in different vascular locations: insights from mouse genetic studies.
This is a review article discussing genetic data from mouse models of atherosclerosis. Some of the mouse strains are well-known in the field, while others are less common. The difference between atherosclerosis susceptibility is discussed on a genetic level looking at mouse QTLs for location specific insights. Specific genes are discussed and linked with the human GWAS data where possible. Discussion is provided looking at the etiology of location-specific susceptibility. Finally, the authors highlight the developmental origin and “memory” of cells within the aorta and it’s major branches, the plasticity that is being uncovered from lineage tracing experiments, and the emerging discovery of substantial heterogeneity based on transcriptomics in these mouse models.
Comments:
The presentation is clear and organized. Statements are well-supported with relevant literature.
Major:
- The manuscript would be stronger if a “caveats” or “limitations” paragraph was added to each section. Likewise, comments on the next steps required to advance the field being discussed should be included in the discussion. As this is entitled “insights”, it should be equally important to acknowledge that mouse models are a very useful tool, but as we’ve seen with drug development for cardiovascular diseases, they have significant limitations.
- In the introduction, the authors describe an outdated mode of atherosclerosis development. There is significant work that has determined it is not endothelial permeability per se, but rather LDL transcytosis that garners LDL access to the subendothelial space in early plaque development. (e.g. Dr. Warren Lee’s group at University of Toronto). Endothelial HMGB1 Is a Critical Regulator of LDL Transcytosis via an SREBP2–SR-BI Axis ATVB 2020.
- Given the insights and conclusions being drawn on the different mouse strains, there should be a comment on whether the literature cited ensured all mouse colonies were clear of any infection between strains and cohorts. Even a small variation with subclinical infection might shift intestinal flora or plaque development.
- The authors should comment on what insights can really be drawn from single cell studies where 60 cells are examined.
- Finally, one macroscopic (and microscopic) difference between atherosclerotic plaques from different locations in humans is the degree of calcification. There should be a section addressing how the mouse models can contribute (or not) to this major topic within atherosclerotic research.
- A final section should be added that discusses insights from mouse models on plaque rupture at different locations. There are relatively few models that mimic the human “vulnerable plaque”. This is a key issue for the field and should be discussed as it limits application to human consequences of atherosclerosis, namely myocardial infarction and stroke.
